# Assessing Red Fox (*Vulpes vulpes*) Demographics to Monitor Wildlife Diseases: A Spotlight on *Echinococcus multilocularis*

**DOI:** 10.3390/pathogens12010060

**Published:** 2022-12-30

**Authors:** Roberto Celva, Barbara Crestanello, Federica Obber, Debora Dellamaria, Karin Trevisiol, Marco Bregoli, Lucia Cenni, Andreas Agreiter, Patrizia Danesi, Heidi Christine Hauffe, Carlo Vittorio Citterio

**Affiliations:** 1Istituto Zooprofilattico Sperimentale delle Venezie, 35020 Legnaro, PD, Italy; 2Conservation Genomics Research Unit, Centre for Research and Innovation, Fondazione Edmund Mach, 38098 San Michele all’Adige, TN, Italy; 3Ethology Unit, Department of Biology, University of Pisa, 56126 Pisa, PI, Italy; 4Wildlife Management Office, Province of Bolzano, 39100 Bolzano, BZ, Italy

**Keywords:** red fox, *Echinococcus multilocularis*, spatial capture-recapture, density, abundance, IKA, non-invasive genotyping, disease surveillance, zoonoses, wildlife management

## Abstract

The assessment of red fox population density is considered relevant to the surveillance of zoonotic agents vectored by this species. However, density is difficult to estimate reliably, since the ecological plasticity and elusive behavior of this carnivore hinder classic methods of inference. In this study, red fox population density was estimated using a non-invasive molecular spatial capture-recapture (SCR) approach in two study areas: one in a known hotspot of the zoonotic cestode *Echinococcus multilocularis*, and another naïve to the parasite. Parasitological investigations on collected samples confirmed the presence of the parasite exclusively in the former area; the SCR results indicated a higher fox population density in the control area than in the hotspot, suggesting either that the relationship between fox density and parasite prevalence is not linear and/or the existence of other latent factors supporting the parasitic cycle in the known focus. In addition, fox spotlight count data for the two study areas were used to estimate the index of kilometric abundance (IKA). Although this method is cheaper and less time-consuming than SCR, IKA values were the highest in the areas with the lower molecular SCR density estimates, confirming that IKA should be regarded as a relative index only.

## 1. Introduction

The red fox (*Vulpes vulpes* L., 1758) is a widespread, synanthropic mesocarnivore, with prevalently nocturnal activity patterns and opportunistic dietary habits [1]. It is also a known reservoir of various pathogens, including zoonotic agents such as the rabies virus [2], *Trichinella* spp. [3], and *Echinococcus* spp. [4]. Understanding fox populations is relevant for both disease surveillance and management; however, estimates of even basic parameters such as density are difficult to obtain, since the ecological plasticity of this species hampers traditional census methods and inference procedures [5,6,7]. Estimates of fox population density from literature vary widely, both in inference procedures and results, ranging from less than one fox per square kilometer [7,8,9], to 1–5 foxes per square kilometer [8,10,11,12] and above [13,14], up to more than 30 foxes per square kilometer [14,15,16,17]. In northeastern Italy, a first step towards standardized data collection was initiated during the most recent rabies epizootic, when demographic trends were inferred by counting foxes during red deer spotlight censuses to estimate the index of kilometric abundance (IKA), with the aim to monitor fox population dynamics, and to model the expected spread and impact of fox-transmitted pathogens [18]. Despite the bias of deer transects towards continuous, open environments, leading to the exclusion of a variety of ecologic niches inhabited by foxes, IKA provided reliable indications of important shifts in fox population density caused by rabies [18]. However, spotlight censuses are not suitable for detecting small demographic effects of milder outbreaks, or to pinpoint host/pathogen interactions that are rarely lethal or cause no clinical signs in the host. In addition, IKA is regarded as a site-specific relative index, and unless a suitable correction is applied to account for variation in local features (e.g. road networks, slope, visibility, etc.) [19], IKA results cannot be extended to other populations and, consequently, to hotspots and naïve areas where the spatial distribution of a host and/or a pathogen is patchy. Such a patchy spatial distribution is a typical feature of *Echinococcus multilocularis* Leuckart, 1863 (Cestoda, Cyclophyllidea, Taeniidae) [20,21,22,23], a small zoonotic tapeworm characterized by an indirect life cycle involving a carnivore definitive host (in subarctic Europe mainly represented by the red fox), and small rodent intermediate hosts [24]. In humans, which are accidental hosts through the exposure to embryonated eggs in feces, *E. multilocularis* is the causative agent of a severe, tumor-like pathology affecting the liver, the alveolar echinococcosis (AE) [4], which is subject to mandatory monitoring in the EU under Directive 2003/99/EC and Regulation (EU) 2016/429. Currently, EU Member States are not obliged to differentially report clinical cases of human AE, and cases of human cystic echinococcosis, which is caused by the congeneric, but ecologically and pathologically distinct *E. granulosus* [4]: thus, in a view to optimize specific *E. multilocularis* monitoring efforts, Member States are encouraged to voluntarily boost the coordination between wildlife pathology experts and human health practitioners [22,25].

In endemic areas, the strategies for preventing human AE strongly rely on individual practices, such as enhancing common hygienic procedures and deworming of domestic carnivores [22,26]; for these measures to become as common as possible, proactive information campaigns to raise awareness of *E. multilocularis* in the exposed populations are highly recommended [22,26,27]. At the same time, scientific research should address the ecological aspects of this pathogen, in order to identify the main factors supporting its persistence, and to determine the strategies to modulate them [22].

In Italy, the distribution of *E. multilocularis* has long been limited to the Alps, corresponding to the southern margin of its central European range. In particular, since 2002, it has consistently been reported in the Alto Isarco-Wipptal district and in Val Pusteria-Pustertal Valley, both in the province of Bolzano-Bozen (northeastern Italy) [21,28,29,30,31,32,33], although recent findings in the Ligurian Alps suggest the emergence of a novel endemic area, possibly as the result of an expansion of its French range towards the southeast [34]. 

Precise information regarding fox population status and dynamics is necessary inside *E. multilocularis* hotspots, as it is crucial to calculate the definitive host sample size needed for monitoring [23], as well as to model human exposure to ensure a risk-based public health surveillance [22]. During recent decades, spatial capture-recapture (SCR) methods have been used to infer population density for a variety of rare, elusive species, e.g. [35,36,37], including the red fox [9,38]. In the present study, molecular SCR was used to estimate fox population density using a non-invasive protocol based on fresh fecal samples. Two areas were investigated and compared: one is a known *E. multilocularis* hotspot, while the other area is naïve to the parasite. The study addressed the hypothesis of a positive relationship between fox population density and the presence of *E. multilocularis*. A secondary aim was to compare IKA and fox population density estimates through SCR, to further evaluate IKA’s possible use and its sensitivity as an index of fox abundance.

## 2. Materials and Methods

The two study areas were located in Alto Isarco-Wipptal (46°52′49.66″ E, 11°20′31.92″ N) and in Val d’Ultimo-Ultental (46°31′18.77″ E, 10°54′35.82″ N), both in the province of Bolzano-Bozen, northeastern Italy (Figure 1). Fox fecal samples were collected in two sampling sessions between June–July 2018 (Alto Isarco-Wipptal) and July–August 2019 (Val d’Ultimo-Ultental).

Scat sampling was carried out following a standardized SCR design [40,41]. Fresh feces were actively sought by walking along an array of parallel transects, 100 m apart, using a GPS device. To account for lower baseline scat detectability in presence of dense understory, the variation in soil visibility was recorded and categorized in three classes (1 = good, 2 = average, 3 = poor) every 100 m and at scat location along each transect. The transects were grouped into discrete zones that could be safely covered by an operator in one working day each. In both study areas, each sampling zone was thoroughly searched during three sampling occasions, with a seven-day interval between each occasion. The Alto Isarco-Wipptal sampling area included eight sampling zones, for a total of 83.2 km of walked transects, oriented along the main geographical axes (i.e. N-S, E-W) (Figure 2). The Val d’Ultimo-Ultental sampling area included five zones, for a total of 42.36 km of transects. Given the presence of higher slopes, transects were oriented approximately in parallel to the main contour lines (Figure 3).

Sampling activities were undertaken by one or two field operators, whose personal safety was monitored remotely through the tracking mobile app Life 360 (Life360 Inc., San Francisco, CA, USA). To avoid bias in detection probability, the same operator provided scat detection throughout both sampling sessions, while a second operator assisted in geo-referencing, data registration, and samples collection. Fox feces were identified based on size, shape, and content [42]; once located, each fresh scat was georeferenced and assigned a unique ID, subdivided into two halves (one for parasitological analysis, one for fox genotyping) and placed in sterile 50 mL Falcon tubes using sterile disposable equipment. Neoprene gloves and an FFP3 mask were worn to prevent fieldworkers from contacting, ingesting, or inhaling *Echinococcus* eggs and to avoid cross-contamination of samples. Samples were carried in a dedicated backpack compartment at ambient temperature and stored at −20 °C within 6 h, until transport to the Conservation Genomics Research Unit at Fondazione Edmund Mach (FEM), where they were kept at −80 °C for at least 72 h to inactivate *Echinococcus* eggs.

Parasitological analyses on collected samples were performed at the Parasitology laboratory of Istituto Zooprofilattico Sperimentale delle Venezie (IZSVe) using the copromicroscopy and multiplex PCR technique described in Citterio et al. [21] and referred to as CMPCR in Obber et al. [33]. Briefly, floatation, filtration, and sieving of 2 g of feces were used to isolate *Taenidae* eggs, which were then processed for DNA extraction, PCR amplification, and Sanger sequencing. True prevalence based on test performances [33] was then calculated using the Rogan–Gladen formula as in Lang and Reiczigel [43].

Fox genotyping took place at the FEM Animal, Environmental, and Antique DNA Platform. For downstream genetic analyses, fecal samples were partitioned under a biological hood in a BSL2 facility dedicated to potentially contaminated material. 200 mg of stool sample were processed for DNA extraction, using E.Z.N.A.^®^ Stool DNA-Human DNA Detection Protocol kit (Omega Bio-tek, Norcross, GA, USA). Extraction products (apx. 130 µL) were frozen as stock, keeping a 30 µL aliquot in refrigerator (4 °C) for subsequent analyses. PCR amplifications were carried out using the marker panel and protocols described in Zecchin et al. [44], with slight modifications to the thermic profiles to enhance performances on a non-invasive matrix (Appendix A). DNA amplicons were genotyped using ABI 3130 XL DNA sequencer and 2.1% GS500LIZ size standard (Applied Biosystems™, Foster City, CA, USA). To avoid contamination, DNA extraction, PCR amplification, and post-PCR operations were carried out in separate laboratories, using dedicated equipment and safety cabinets; negative controls were used at every step to exclude contamination. To minimize the effects of allelic dropout and null alleles, each sample was amplified using a multiple tube approach [45], thus providing at least two consensus repetitions for heterozygote alleles, and at least four for homozygotes. Fragment size was scored using GeneMapper^®^ 5.0 (Applied Biosystems^TM^, Foster City, CA, USA), considering the successful amplification of at least nine loci as a threshold for reliable individual characterization. GenAlEx 6.5 [46] was used to identify unique genotypes, allowing for missing data to match with otherwise identical profiles, and to compute the main genetic diversity indices. Genotyping success and null allele frequencies were assessed using MICRO-CHECKER [47].

To infer the abundance (N) of the fox populations, the corresponding capture histories, representing the localization of each individual’s scat, were analyzed in a Bayesian framework using Markov Chain Monte Carlo (MCMC) algorithm implemented in OpenBUGS 3.2.3 [48,49]. As in classical SCR models, the probability of detecting one individual (in this case, one individual’s scat) was set to decrease with the distance between each latent individual activity center (s_i_) and the ‘detectors’ array, represented by a grid of virtual points (the ‘detectors’, *trap_j_*) along each transect. The linear predictor was integrated with the parameter expressing soil visibility, as registered during sampling for each point along the transect line (Equation (1)). The two covariates were included as a single predictor variable into Gompertz’s risk function [40,50] (Equation (2)). As an example, the spatial decline in detection probability in zone 5 (Alto Isarco-Wipptal study area) is represented in Figure 4.
(1)$$VV_{i,t,j}=\sqrt{{\left(u_{i,t,x}-trap_{j,x}\right)}^{2}+{\left(u_{i,t,y}-trap_{j,y}\right)}^{2}}*trap_{j,vis}$$
(2)$$\log\left(h_{i,t,j}\right)=-\beta_{0}-\beta_{1}*VV_{i,t,j}$$

As the sampling interval did not overlap with recruitment season and was short enough for the effect of mortality and migration to be considered negligible, fox populations at both sampling sites were regarded as closed within each sampling session. To specify the prior distribution of N, the *n*-sized datasets were artificially inflated through the technique of data augmentation [51,52]. The prior distribution for the movement parameter sigma (σ) was set to a maximum of 1 km, considering that sampling activities took place when fox biological period is characterized by relatively small excursions [53,54]. For both study areas the state-space *S*, representing the surface containing virtually all individual activity centers, and to which density estimates referred to, was defined as a buffer around groups of zones, and identified by MCMC convergence for N starting from different priors, by trial-and-error model runs. OpenBUGS input files and model parameters were specified in R 4.2.0 [55] using R2OpenBUGS package [56]. Three parallel Monte Carlo chains of 55,000 iterations each were run, discarding the first 5000 (burn-in), and then half of the remaining MCMC samples (thinning rate), to reduce autocorrelation and influence of initial values. To further reduce autocorrelation, OpenBUGS over-relaxation option was used. Convergence of Markov chains was assessed through Brooks-Gelman-Rubin (BGR) diagnostics (R < 1.1) and evaluation of Monte Carlo standard error (<5% of the posterior standard deviation—SD) [57]. BUGS model is included in Appendix A.

In order to evaluate a possible relationship between fox population density estimates and IKA, fox spotlight count data for 2018 (Alto Isarco-Wipptal) and 2019 (Val d’Ultimo-Ultental) were analyzed. The data, collected by the Bolzano-Bozen Provincial Forestry Service in collaboration with local hunters, were obtained by counting foxes while driving along fixed road transects during red deer census activities, as described in Obber et al. [18]. Corresponding IKAs were then calculated by dividing the highest fox count registered inside each state-space, during the census season, by the distance driven in kilometers. For both study areas, the factor connecting IKA to the respective density estimate was calculated as IKA/density ratio.

## 3. Results

During the 2018 sampling session (Alto Isarco-Wipptal), 56 fecal samples were collected and genotypes were obtained for 39 (69.6%) samples, corresponding to 31 individual foxes (17 males, 11 females, 3 undetermined); the successfully genotyped samples, as well as baseline detection probability, are represented in Figure 5. According to parasitological investigations, two fecal samples from Alto Isarco-Wipptal, corresponding to two individual foxes (6.45%), tested positive for *E. multilocularis*. Based on test performances according to Obber et al. [33], true prevalence was estimated at 31% (CI 50%: 12–43.5).

The 2019 sampling session (Val d’Ultimo-Ultental) yielded 59 fecal samples, and 31 valid genotypes (52.5%) were obtained, representing 26 individuals (15 males, 9 females, 2 undetermined); the successfully genotyped samples, as well as baseline detection probability, are represented in Figure 6. After CMPCR, all samples collected in Val d’Ultimo-Ultental tested negative for *E. multilocularis*. Alto Isarco-Wipptal and Val d’Ultimo-Ultental capture histories are included as Appendix A, respectively.

The main genetic diversity indices were similar for both fox populations: for Alto Isarco-Wipptal and Val d’Ultimo-Ultental sampling sites, respectively, the marker panel revealed a mean of 8.05 ± 0.47 and 7.25 ± 0.37 alleles per locus, and in both cases, F_is_ values (−0.014 and −0.009) did not suggest significant deviations from panmixia. For the Alto Isarco-Wipptal population only, analysis with MICROCHECKER highlighted the possible presence of null alleles at locus FH2001 (estimated null allele frequency 0.091).

In Alto Isarco-Wipptal, four individual foxes were sampled twice. After OpenBUGS trial runs, convergence of Markov chains was achieved considering a 112 km^2^ state-space (*S*), which encompassed all eight zones within a one km buffer (Figure 5). Inside *S*, population abundance was estimated at 319 ± 72.6 (CI 95%: 199–476) individuals, resulting in a density estimate of 2.97 ± 0.64 foxes/km^2^. BGR ratio fell under the 1.2 threshold almost immediately after burn-in, and final Monte Carlo error (2.32% of the posterior standard deviation) was considered acceptable. The analysis took approximately 10 days on a standard office PC. Fox spotlight count data, represented in Figure 7, yielded the highest IKA on July 12, 2018 (0.51 foxes/km), corresponding to a density/IKA ratio of 5.82 ± 1.25.

In Val d’Ultimo-Ultental, four foxes were sampled twice, and one fox three times. After trial and error model runs, Markov chains convergence was achieved considering two different state-spaces (*S*A and *S*B), identified around zones 1, 2, 3 (one km buffer, 40.8 km^2^) and 4, 5 (0.5 km buffer, 15.2 km^2^), respectively (Figure 6). Fox population abundance estimate for *S*A was 144.7 ± 39 (CI 95%: 81–232) individuals, corresponding to 3.55 ± 0.96 foxes/km^2^; after the burn-in, BGR ratio never rose above 1.2, and final Monte Carlo error was 1.89% of the posterior standard deviation. The analysis took approximately 5 days on a standard office PC. In *S*B, fox population was estimated as 101.3 ± 25.8 (CI 95%: 60–160) individuals, or 6.66 ± 1.7 foxes/km^2^; as in SA, BGR ratio never rose above 1.2 after the burn-in, and final Monte Carlo error was 1.65% of the posterior standard deviation. The analysis took approximately 27 h. To account for the overall Val d’Ultimo-Ultental study area, fox population density was calculated as the mean of the two state-spaces at 5.11 ± 0.97 foxes/km^2^. IKA was registered on 29 March 2019 (0.18 foxes/km), corresponding to a density/IKA ratio of 29.5 ± 5.72. Fox spotlight count data are represented in Figure 8.

## 4. Discussion

A positive relationship between host population density and parasitic abundance, as comprehensive of both burden within hosts and prevalence in the host population [58], by means of leveraging transmission rate, is a crucial assumption in epidemiology [59,60], and is currently widely applied in modeling disease transmission and persistence (see [61] for a review). On the other hand, fox population density as measured by SCR in the present study was much lower inside the known *E. multilocularis* hotspot of Alto Isarco-Wipptal (2.97 ± 0.64 foxes/km^2^), compared to the naïve territory of Val d’Ultimo-Ultental (5.11 ± 0.97 foxes/km^2^). The parasitological analyses carried out on collected samples confirmed the former area to be a highly endemic *E. multilocularis* hotspot, and the latter as a naïve territory [21,33]. However, the sample size here considered, as well as the test sensitivity, were too low for any robust inference on the true prevalence. Therefore, the value of 31% prevalence for Alto Isarco-Wipptal should be considered very cautiously until further verification by ad hoc sampling and sensitive diagnostic tools, e.g. qPCR [33]. Although these results do not rule out fox density as an influencing factor for the persistence of the parasite in the Alto Isarco-Wipptal hotspot, and consequently for human exposure, they suggest either a nonlinear relationship between fox density and parasite prevalence, and/or the existence of other latent factors supporting the parasitic focus. In fact, many studies have reported a deviation from the assumption of a linear host density–pathogen relationship (e.g. [58,62,63]), concerning in particular indirectly transmitted parasites [58], such as *E. multilocularis*. The results of the present study highlight a lack of knowledge concerning the ecology of this cestode, and encourages further investigations to disentangle and possibly model the dynamics between foxes, small rodents, and *E. multilocularis*, and therefore to address possible disease control strategies [61,63]. In particular, the abundance and structure of the small mammals’ intermediate host community are likely to play an important role for the persistence of the zoonotic hotspots and should be targeted by future research.

As another reason for the apparent lack of association between fox density and parasite presence, foxes respond both spatially and demographically to changes in environmental carrying capacity, which depends on food abundance [64]. In highly urbanized contexts, fox communities have shown lower parasitic prevalence compared to those inhabiting the rural outskirts, possibly due to scantier intermediate hosts communities pushing foxes towards anthropogenic, non-infective food sources [65,66]. Although the highly fragmented, rural Alpine environments will likely provide foxes with natural food supply, also in proximity to human settlements, future research should include studies of prevalence of *E. multilocularis* in rodent intermediate hosts to understand the factors driving these hotspots.

While it has been suggested that numeric control of fox population could lower the environmental contamination with *E. multilocularis* eggs [67], the eradication of the parasite by this means was achieved only once in a small, closed population, and it is considered unfeasible for large areas [22,26,68]. The results presented here corroborate how extinguishing the parasitic focus by fox culling would be untenable, given that the hotspot of Alto Isarco-Wipptal already had a low density of the definitive host population, compared to the naïve area of Val d’Ultimo-Ultental. In fact, depopulation would likely increase the probability of importing the parasite from nearby foci by enhancing fox movement and inducing dispersal from outer populations through compensatory migration dynamics [69,70]. On the other hand, sound waste disposal measures in semi-urbanized contexts has proven effective in lowering fox numbers, while at the same time inducing an expansion of individual activity centers [71], thus reducing the effect of compensatory migrations by means of extending territorial activity. Moreover, in rural areas such as those considered in the present study, which are characterized by a complex and diverse land use patterns, small rodent communities are often ubiquitous, and cost/benefit ratio is unlikely to favor the application of specific management strategies of intermediate hosts, e.g. [72].

The increase in fox numbers has been linked to *E. multilocularis* emergence in Europe [23,73] and, together with fox urbanization, to a surge in human alveolar echinococcosis cases [26,74,75]. Recently, the fox population of northeastern Italy has been affected by two epizootics: an unprecedented canine distemper wave in 2006, arriving from countries on Italy’s northeastern borders [44,76], followed by rabies from the Balkans during 2008–2011 [77,78]. Rabies was eradicated by intensive oral vaccination campaigns, and its spread halted before reaching the study areas described here [78,79]. Conversely, canine distemper became endemic across the Alps, showing different variants as well as a complex geographical pattern of cyclical epizootics [80]. The rapid recovery from highly lethal epizootics has likely contributed to the rise in fox numbers, as well as to the recolonization of empty territories, as reported elsewhere [81,82,83,84,85]. While IKA proved sensitive enough to detect such high lethality [18], the present study confirms that it should be regarded as a relative index only, given that the highest value came from the study site with the lowest abundance of foxes as measured by SCR. In order to draw a regression model between IKA and population density at a local level, repeated measurements of both IKA and population density would be required.

## 5. Conclusions

As mentioned in the Introduction, and considering how intervening on the ecological cycle of the parasite would be unlikely to lead to successful outcomes against human exposure to *E. multilocularis*, public health measures should focus primarily on: (I) informing the local human population, starting from individuals at particularly high risk of exposure, such as hunters and game wardens; (II) initiating an exchange of information among the stakeholders (not only wildlife managers and veterinary services, but also general and specialist practitioners); (III) targeting surveillance in foxes to detect new endemic areas, while collecting all data relevant for risk assessment. Long-term data on trends in fox density would be particularly valuable for priority; (IV) using a robust approach as outlined in this study. Moreover, future insights in the eco-epidemiology of *E. multilocularis* should carefully consider factors other than the definitive host, including the dynamics of the intermediate host populations, as well as the parasite’s presence in these species, with a particular focus on those known to be highly susceptible to this cestode, such as *Arvicola* spp. and *Microtus* spp. [64]. Due to the patchy distribution of *E. multilocularis*, as well as to the complexity of its life cycle, such investigations should be performed at a small scale, as already reported for parasite detection in apparently naïve areas [33].

## Figures and Tables

**Figure 1 pathogens-12-00060-f001:**
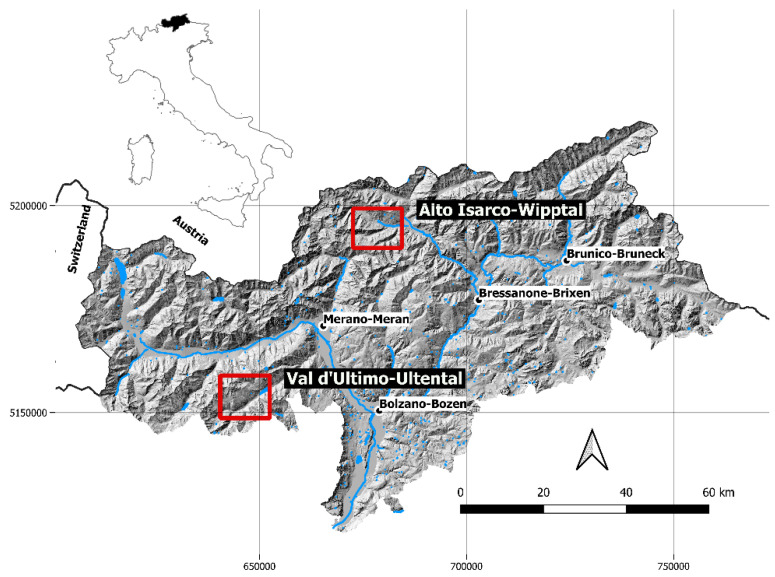
Location of the two study areas of Alto Isarco-Wipptal and Val d’Ultimo-Ultental (red squares) in the province of Bolzano-Bozen, northeastern Italy. The two sampling sessions were carried out in 2018 and 2019, respectively. Hillshade based on TINITALY digital elevation model (CC BY 4.0) [39] was used as a base layer.

**Figure 2 pathogens-12-00060-f002:**
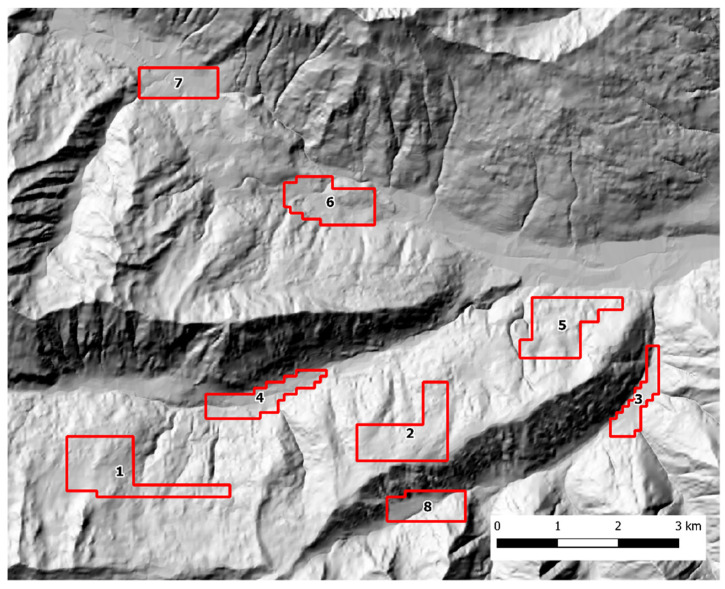
The Alto Isarco-Wipptal study area. Sampling zones 1 to 8 are indicated. Hillshade based on TINITALY digital elevation model (CC BY 4.0) [39] was used as base layer.

**Figure 3 pathogens-12-00060-f003:**
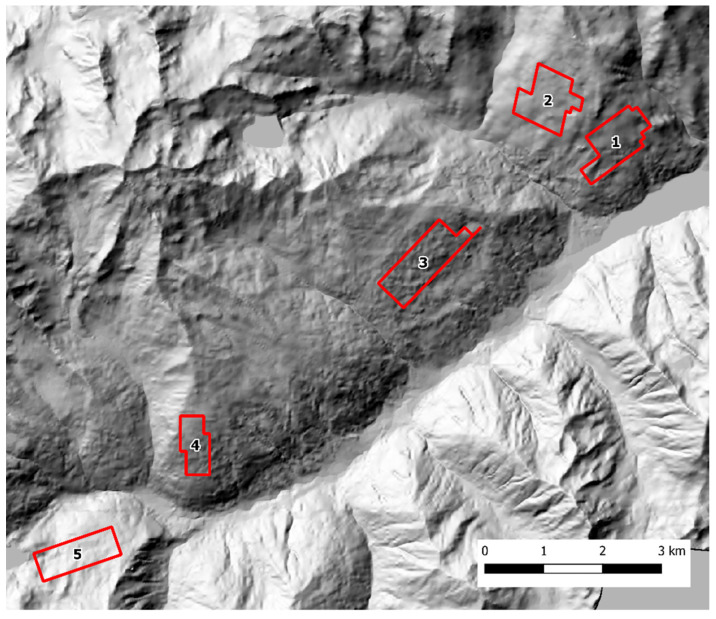
The Val d’Ultimo-Ultental study area. Sampling zones 1 to 5 are indicated. Hillshade based on TINITALY digital elevation model (CC BY 4.0) [39] was used as base layer.

**Figure 4 pathogens-12-00060-f004:**
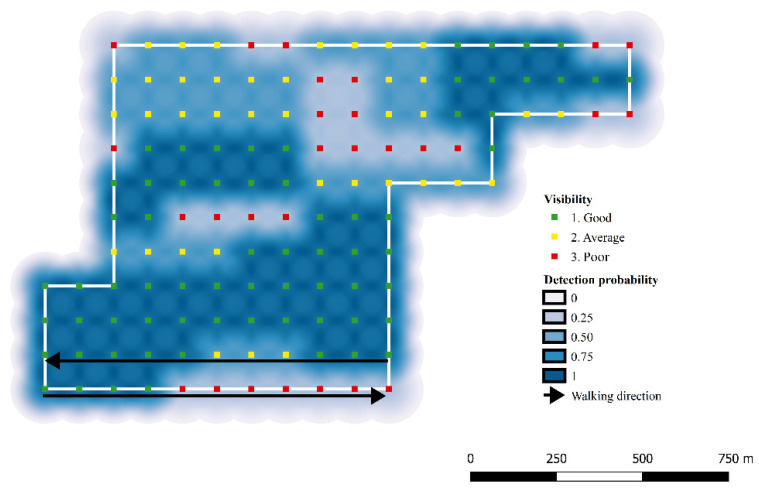
Example of decline in detection probability of red fox as a function of distance and baseline visibility: zone 5, Alto Isarco-Wipptal study area.

**Figure 5 pathogens-12-00060-f005:**
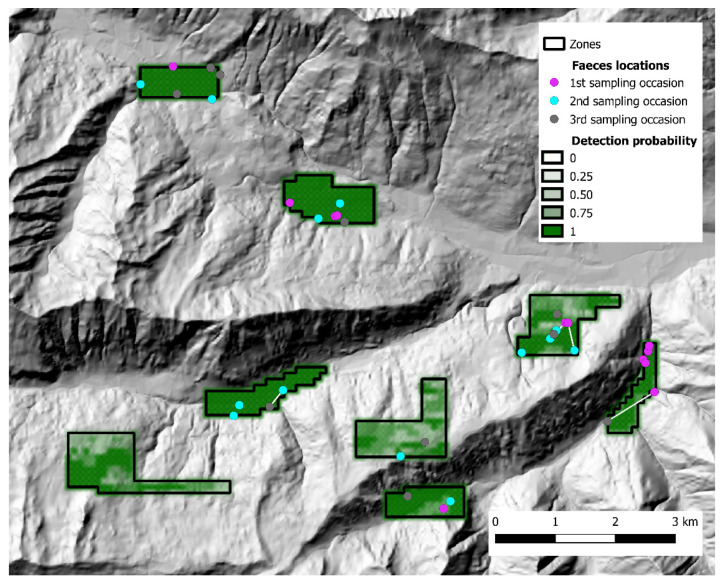
State-space and distribution of fecal samples collected in Alto Isarco-Wipptal study area. Samples assigned to the same individual fox are connected by a white line. Baseline detection probability inside each zone is represented in shades of green. Hillshade based on TINITALY digital elevation model (CC BY 4.0) [39] was used as base layer.

**Figure 6 pathogens-12-00060-f006:**
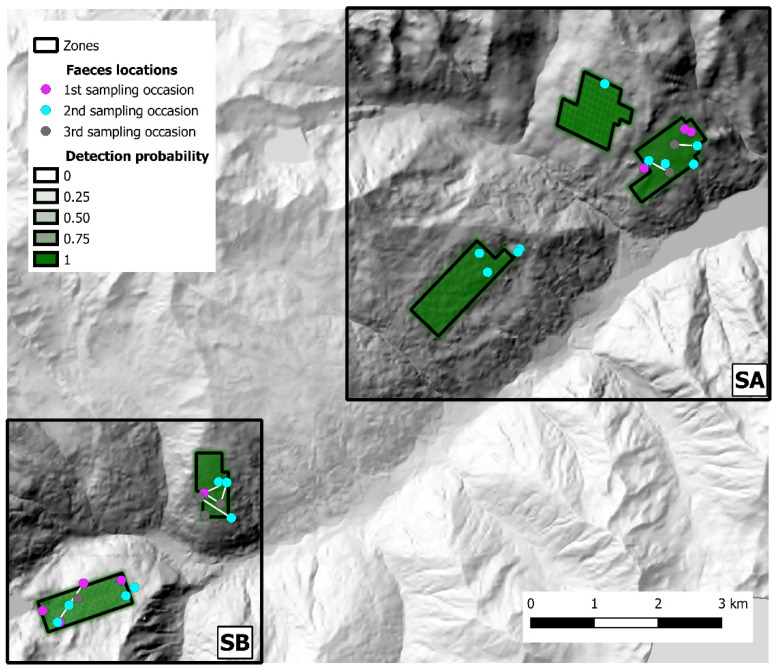
State-spaces (*S*A and *S*B) and distribution of fecal samples collected in Val d’Ultimo-Ultental study area. Samples assigned to the same individual fox are connected by a white line. Baseline detection probability inside each zone is represented in shades of green. Hillshade based on TINITALY digital elevation model (CC BY 4.0) [39] was used as base layer.

**Figure 7 pathogens-12-00060-f007:**
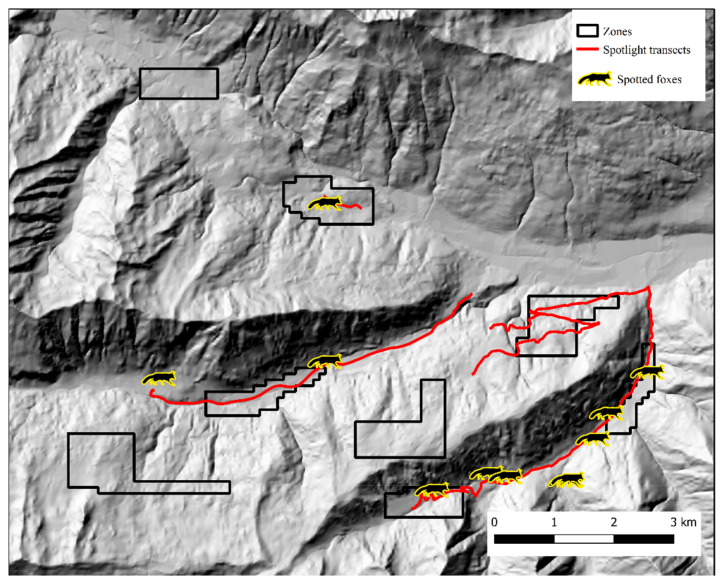
Spotlight routes (red) and fox counts (silhouettes) in Alto Isarco-Wipptal study area. Hillshade based on TINITALY digital elevation model (CC BY 4.0) [39] was used as base layer.

**Figure 8 pathogens-12-00060-f008:**
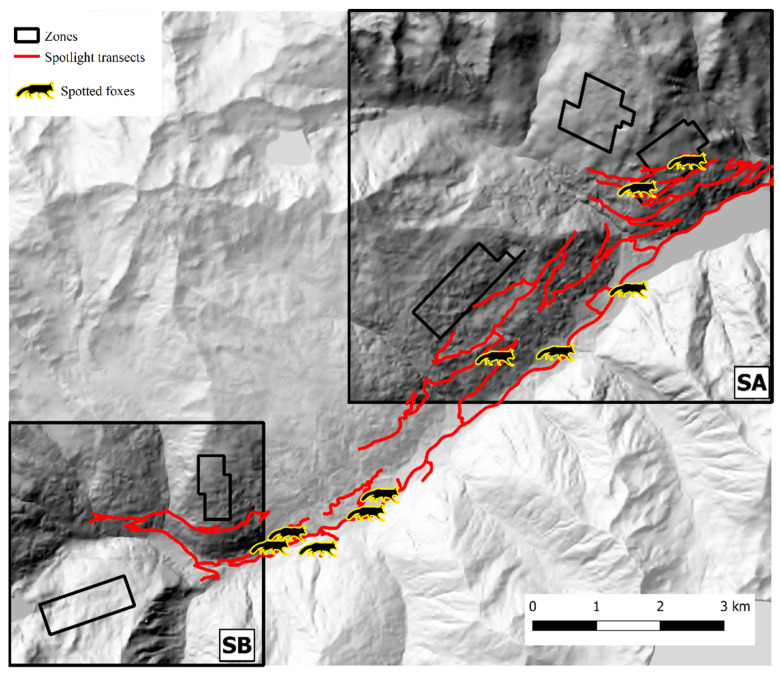
Spotlight routes (red) and fox counts (silhouettes) in Val d’Ultimo-Ultental study area. Hillshade based on TINITALY digital elevation model (CC BY 4.0) [39] was used as base layer.

## Data Availability

All relevant data are included within the manuscript and its Appendix A.

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
