# Peer review of "Assessing Red Fox (Vulpes vulpes) Demographics to Monitor Wildlife Diseases: A Spotlight on Echinococcus multilocularis"

_pathogens, 2022, doi:10.3390/pathogens12010060_

Round 1

Reviewer 1 Report

I found the manuscript well written and for most part clear. The study design and analysis for SCR are overall strong and clearly explained. I have one major comment regarding how this manuscript was framed that I think the authors should carefully evaluate.

Because of the title “spotlight n E. multilocularis” and the entire introduction (and discussion), I was very surprised when at the end of Materials and methods I realized that no parasitological analyses were performed on the faecal samples. A paper by the same research group Obber et al 2022 shows how faecal samples can be used to detect the parasite and assess prevalence though either qPCR or CMPCR, so I do not understand why the samples were not analyzed with the same approach. The authors claim that the 2 study areas were selected because one is a verified hot spot for the parasite but the other is naïve to the parasite. While I see in literature evidence for the hot-spot, I could not find anything about the latter and a reference is not reported in the manuscript. The number of samples collected would have been enough to report prevalence of the parasite in those sample areas. Instead, with that part missing, the paper becomes more like an exercise on the application of SCR methods on scat samples, approach that has already been validated for several species in many habitats and for many carnivore/mesocarnivore species using scats. That takes to a discussion that is completely speculative because it goes back to the parasite and wildlife disease monitoring and surveillance without any real new information about it other than that the population density can be estimated through that approach. Moreover, the discussion starts with the concept of a positive relationship between host population density and parasitic abundance that is, as stated, a crucial assumption in epidemiology. However, it cannot be determined by the analyses carried out since there are so many other factors that can affect the population density other than parasites, most of which are mentioned in the discussion, but none of which are investigated in the study. So in my opinion no conclusions can be drawn from the density of the fox population in relation to E. multilocularis because, other than several ecological factors, even other parasites could be responsible for different densities in those study areas. I highly suggest the authors to completely reframe the manuscript to “we want to test this study design and analytical approach to see if it could be implemented in any type of wildlife disease monitoring in Italy” (where parasites can be detected in the faeces) instead of focusing on one specific parasite. In this case, I do not think the journal selected would be appropriate.

Please find my minor comments in the file attached.

Reviewer 2 Report

The manuscript assessed red fox population density in the Italian Alps using non-invasive genotyping and Bayesian modeling from the viewpoint of zoonotic diseases. Specifically, density estimates were compared between a known hotspot of Echinococcus multilocularis, and another naive to the parasite. The main finding is that fox population density was much lower inside the known E. multilocularis hotspot than in the naive area, suggesting that the presence of the parasite is not related to the population density of the red fox.

The manuscript is deliberate and mainly well-written. The figures are clear, but there are some issues with numbering. The fourth figure (in line 151) is denoted erroneously as Figure 1, and Figure 7 is mentioned in the text before Figure 6, the numbering might be changed.

It is mentioned in line 126 that the PCR protocol was slightly changed, but the changes are not detailed. However, the applied modifications should be mentioned here.

Consider rephrasing the sentence starting in line 131 to something like: „ Fragment sizes were scored using Gene-Mapper…”.

You should be consistent with indicating the version number of the software used. If you indicate it by OpenBUGS in line 140, you should also indicate it by R in line 162.

The genotyping success was 69.6% and 52.5% in the two sampling sites, which I find somewhat low. Especially when „amplification of at least nine loci as a threshold” (out of 21 markers) was considered successful genotyping. Did you try to re-amplify the failed samples? You should be clear about this in the methods section.

The abbreviation BGR in line 209 is not resolved in the text. Additionally, in lines 219 and 222 the abbreviation reads as BRG. Please correct this.

The parentheses around the in-text citations in lines 238 and 247 are superfluous, they could be simply omitted.

Reviewer 3 Report

Please refer to the attached pdf.
